

# The non-rational limit of D-series minimal models

**Sylvain Ribault**⋆

Institut de Physique Théorique, Université Paris-Saclay, CEA, CNRS

⋆ sylvain.ribault@ipht.fr

## Abstract

We study the limit of D-series minimal models when the central charge tends to a generic irrational value $c \in (-\infty, 1)$. We find that the limit theory's diagonal three-point structure constant differs from that of Liouville theory by a distribution factor, which is given by a divergent Verlinde formula. Nevertheless, correlation functions that involve both non-diagonal and diagonal fields are smooth functions of the diagonal fields' conformal dimensions. The limit theory is a non-trivial example of a non-diagonal, non-rational, solved two-dimensional conformal field theory.

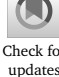
# 1 Introduction

The exploration of two-dimensional conformal field theories has begun with theories that involve finitely many irreducible representations of the Virasoro algebra. Such theories have been classified, and they are called minimal models [1]. Minimal models can be exactly solved, and some of them describe interesting physical systems such as the critical Ising model. However, minimal models only exist for rational values of the Virasoro algebra's central charge, while interesting conformally invariant systems can have more general central charges: for the critical $Q$-state Potts model [2] or Liouville theory [3], the central charge $c$ is a continuous parameter.

Algebraically, minimal models look very different from generic $c$ theories: the former involve finitely many intricate representations, while the latter involve infinitely many simple representations, and are therefore non-rational. However, at the level of correlation functions, the difference is not that sharp: correlation functions of minimal models not only are special values of generic $c$ expressions [4], but also can have well-defined limits when the central charge tends to irrational values [3, 5]. In this article, we will use a limit of D-series minimal models for building and solving a non-diagonal theory for generic $c \in (-\infty, 1)$.

But why use such a complicated approach? Why not use known analytic bootstrap techniques, based on the two assumptions that degenerate fields exist and that correlation functions depend analytically on conformal dimensions? After being introduced in the context of Liouville theory [6], these techniques were recently extended to the case of non-diagonal theories [5], leading to equations that determine how correlation functions depend on the fields' conformal dimensions. However, in the presence of non-diagonal fields, these equations do not have solutions that are analytic in the diagonal fields' conformal dimensions, as we will review in Section 2.2. The solution for the three-point structure constant of diagonal fields will not even be a function of the conformal dimensions, but a distribution. (See Section 4.3 for its expression and properties.) Taking limits of known minimal model expressions is a way to compute this distribution.

Taking limits however comes with its own subtleties: most notably, the limits of minimal models' correlation functions can belong to two different theories, depending on whether non-diagonal fields are present. While correlation functions of diagonal fields plainly tend to correlation functions of Liouville theory, the limits of correlation functions that involve nondiagonal fields do not belong to some extension of Liouville theory. Rather, they belong to a theory whose diagonal sector differs from Liouville theory, and whose diagonal three-point structure constant is a distribution. In other words, **the diagonal sector of the limit theory differs from the limit of the diagonal sector**. We will discuss the mechanism for this difference in Section 5.

Let us sketch the techniques that we will use. Correlation functions of minimal models can be decomposed into sums of finitely many conformal blocks, but the number of blocks tends to infinity when $c$ tends to an irrational value. This leads to sums over infinite sets of the type $\beta \mathbb{Z} + \beta^{-1} \mathbb{Z}$, which we will call squashed lattices. (See Eq. (2.3) for the relation between $c$ and $\beta$.) We will find that such sums can be rewritten as integrals, see the mathematical interlude Section 3. This rewriting works provided $\beta^2$ is irrational, and also obeys number-theoretic assumptions on its Diophantine approximations, which however only exclude a set of values of measure zero. Applying these results to correlation functions, we find that the limit of minimal models exists for generic values of the central charge. We will also provide independent checks by numerically testing crossing symmetry in Section 4, which will confirm that we obtain a consistent CFT. The corresponding Python code is available at GitLab [7].

In order to distinguish our results from previous work, let us emphasize that we find a two-dimensional CFT that is **fully solved** (on the plane), **non-trivial**, **non-diagonal**, and exists for

**generic central charges**. Relaxing any one of these four properties, we would find other examples in previous work:

- Not fully solved: the $Q$-state Potts model [8], the limit of D-series minimal models when it was first proposed [5].

- Trivial: compactified free bosons at arbitrary central charges [3].

- Diagonal: Liouville theory, generalized minimal models [3].

- Rational central charges: D-series and E-series minimal models [1].

Another candidate might be the $SL_2(\mathbb{R})$ WZW model [3], where by $SL_2(\mathbb{R})$ we mean the group and not its universal cover, as taking the universal cover would make the model diagonal [9]. This model however has more than Virasoro symmetry, and is not quite fully solved.

   While our methods work for generic $c \in (-\infty, 1)$, our theory surely exists for $\Re c < 13$, as we will argue in Section 6. The extension from the half-line to the half-plane cannot be done by analytic continuation, and will require other techniques.

## 2 Limit of minimal models: easy bits and tricky bits

In the bootstrap approach to conformal field theory, correlation functions are assembled from three ingredients: the spectrum, structure constants, and conformal blocks. The spectrum and structure constants are model-dependent data, while conformal blocks are universal functions of the fields' positions and conformal dimensions. The consistency of a conformal field theory on the sphere reduces to crossing symmetry of four-point functions, so we will be particularly interested in four-point functions. Crossing symmetry amounts to the agreement of the $s$-, $t$- and $u$-channel decompositions of any given four-point function, schematically:

$$
s \quad = \quad t \quad = \quad u \tag{2.1}
$$

For example, the $s$-channel decomposition of a four-point function reads

$$
\left\langle \prod_{i=1}^{4} V_i(z_i) \right\rangle = \sum_{j \in \mathcal{S}^{(s)}} \frac{C_{12j} C_{j34}}{B_j} \mathcal{F}_j^{(s)}(\{z_i\}) , \tag{2.2}
$$

where

- the $s$-channel spectrum $\mathcal{S}^{(s)}$ is a subset of the spectrum of our CFT, determined by the fusion rules for $V_1 V_2$ and $V_3 V_4$,

- $B_i$ and $C_{ijk}$ are respectively two- and three-point structure constants,

- $\mathcal{F}_j^{(s)}(\{z_i\})$ is a four-point conformal block, which depends not only on the fields' parameters (for example, conformal dimensions) but also on their positions $z_i$.

Taking limits would be straightforward if these ingredients where smooth functions of the central charge and of the fields' conformal dimensions. And to a large extent they are smooth, in particular the conformal blocks are meromorphic when written in terms of the right variables. We will now sketch how the spectrum and structure constants behave in the non-rational limit of minimal models.

## 2.1 Spectrum and fusion rules: easy bits

**Review of the spectrum**

Let us parametrize the central charge $c$ of the Virasoro algebra in terms of a number $\beta$ such that

$$c = 1 - 6\left(\beta - \frac{1}{\beta}\right)^2 . \tag{2.3}$$

Minimal models exist for positive rational values of $\beta^2$ of the type

$$\beta^2 = \frac{p}{q} \qquad \text{with} \qquad p, q \geq 2 \text{ coprime integers .} \tag{2.4}$$

The spectrums of minimal models are built from degenerate highest-weight representations of the Virasoro algebra. For $r, s \in \mathbb{N}^*$, we call $\mathcal{R}_{\langle r,s \rangle}$ the degenerate highest-weight representation whose highest-weight state has the momentum

$$P_{\langle r,s \rangle} = \frac{1}{2}\left(\beta r - \frac{s}{\beta}\right) , \tag{2.5}$$

where the momentum $P$ is related to the conformal dimension $\Delta$ by

$$\Delta = \frac{c-1}{24} + P^2 . \tag{2.6}$$

Following [4], we write the spectrums of D-series minimal model as

$$\mathcal{S}_{p,q}^{\text{D-series}} = \frac{1}{2} \bigoplus_{\substack{(r,s) \in K_{p,q} \\ rs \in \mathbb{Z} + \frac{1}{2} + \frac{pq}{4}}} \left| \mathcal{R}_{\langle r,s \rangle} \right|^2 \oplus \frac{1}{2} \bigoplus_{\substack{(r,s) \in K_{p,q} \\ rs \in \mathbb{Z}}} \mathcal{R}_{\langle r,s \rangle} \otimes \bar{\mathcal{R}}_{\langle -r,s \rangle} , \tag{2.7}$$

where the shifted Kac table is

$$K_{p,q} = \left[ \left(\mathbb{Z} + \tfrac{q}{2}\right) \cap \left(-\tfrac{q}{2}, \tfrac{q}{2}\right) \right] \times \left[ \left(\mathbb{Z} + \tfrac{p}{2}\right) \cap \left(-\tfrac{p}{2}, \tfrac{p}{2}\right) \right] . \tag{2.8}$$

This notation allows indices $r, s \in \frac{1}{2}\mathbb{Z}$ rather than $r, s \in \mathbb{N}^*$. This is possible thanks to the identity $P_{\langle r,s \rangle} = P_{\langle r+\frac{q}{2}, s+\frac{p}{2} \rangle}$, which we interpret as implying $\mathcal{R}_{\langle r,s \rangle} = \mathcal{R}_{\langle r+\frac{q}{2}, s+\frac{p}{2} \rangle}$.

The spectrum $\mathcal{S}_{p,q}^{\text{D-series}}$ is made of two terms, which we will call diagonal and non-diagonal. In the diagonal sector, a representation of the left-moving Virasoro algebra is coupled to the same representation of the right-moving Virasoro algebra. In the non-diagonal sector, the two coupled representations $\mathcal{R}_{\langle r,s \rangle}$ and $\bar{\mathcal{R}}_{\langle -r,s \rangle}$ generically differ, but they happen to coincide if $rs = 0$. For us, a non-diagonal representation is not defined as a representation whose primary state has nonzero conformal spin i.e. $\Delta \neq \bar{\Delta}$. Rather, the non-diagonal sector is defined as the odd sector with respect to the $\mathbb{Z}_2$ symmetry of the model [10].

**Limit of the non-diagonal sector**

Let us consider the behaviour of the spectrum $\mathcal{S}_{p,q}^{\text{D-series}}$ in the limit

$$p, q \to \infty \qquad , \qquad \frac{p}{q} \to \beta^2 \in \mathbb{R}_{>0} - \mathbb{Q} . \tag{2.9}$$

In order to fully characterize the limit, we should specify how the indices $r, s$ behave. We first focus on the non-diagonal sector. This sector is not empty provided one of the integers $p, q$

is even. In this sector, the number $rs = \Delta_{\langle -r,s\rangle} - \Delta_{\langle r,s\rangle}$ has an interpretation as the conformal spin of the representation $\mathcal{R}_{\langle r,s\rangle} \otimes \bar{\mathcal{R}}_{\langle -r,s\rangle}$, so it must remain integer in our limit, and therefore constant. This suggests that each index $r,s$ should be constant. Since $(r,s) \in \left(\mathbb{Z} + \frac{q}{2}\right) \times \left(\mathbb{Z} + \frac{p}{2}\right)$, both integers $p,q$ should have constant parity.

We therefore have two choices: $p$ odd and $q$ even, or the opposite. We choose

$$p \text{ odd and } q \text{ even}, \tag{2.10}$$

at the price of breaking the symmetry $p \leftrightarrow q$ i.e. $\beta \leftrightarrow \beta^{-1}$, a symmetry which manifests itself in the identity $\mathcal{S}_{p,q}^{\text{D-series}} = \mathcal{S}_{q,p}^{\text{D-series}}$. In previous works [4, 5], we considered that we had two different limit CFTs for each value of the central charge. Here, we consider that we have one limit CFT that depends on $\beta^2$ rather than on $c$. The non-diagonal sector of the limit CFT is

$$\lim_{\substack{p,q\to\infty \\ \frac{p}{q}\to\beta^2 \\ (p,q)\in(2\mathbb{N}+1)\times 2\mathbb{N}^*}} \frac{1}{2} \bigoplus_{\substack{(r,s)\in K_{p,q} \\ rs\in\mathbb{Z}}} \mathcal{R}_{\langle r,s\rangle} \otimes \bar{\mathcal{R}}_{\langle -r,s\rangle} = \frac{1}{2} \bigoplus_{(r,s)\in 2\mathbb{Z}\times(\mathbb{Z}+\frac{1}{2})} \mathcal{V}_{P_{\langle r,s\rangle}} \otimes \bar{\mathcal{V}}_{P_{\langle -r,s\rangle}}, \tag{2.11}$$

where the degenerate representations $\mathcal{R}_{\langle r,s\rangle}$ become Verma modules $\mathcal{V}_P$ due to their null vectors escaping to infinite level. In this sector, the left and right momentums belong to a rectangular lattice in $\mathbb{R}^2$,

$$(P,\bar{P}) \in \beta\mathbb{Z}(1,-1) + \frac{1}{2\beta}\mathbb{Z}(1,1) + \frac{1}{4\beta}(1,1), \tag{2.12}$$

and the total conformal dimension takes discrete values that are not dense and reach $+\infty$,

$$\Delta_{\langle r,s\rangle} + \Delta_{\langle -r,s\rangle} = \frac{c-1}{12} + \frac{1}{2}\left(\beta^2 r^2 + \beta^{-2} s^2\right). \tag{2.13}$$

In particular, the lowest total dimension in the non-diagonal sector is $2\Delta_{\langle 0,\frac{1}{2}\rangle} = \frac{c-1}{12} + \frac{1}{8\beta^2}$. This is not invariant under $\beta \to \beta^{-1}$, which illustrates the fact that the two CFTs with parameters $\beta$ and $\frac{1}{\beta}$ are different if $\beta^2 \neq 1$.

**Limit of the diagonal sector**

In the diagonal sector, i.e. the first term of $\mathcal{S}_{p,q}^{\text{D-series}}$, the representation $\left|\mathcal{R}_{\langle r,s\rangle}\right|^2$ is characterized by the momentum $P_{\langle r,s\rangle}$. In our limit (2.9), the momentums $P_{\langle r,s\rangle}$ that appear in the diagonal sector become uniformly distributed on the real line. The uniform distribution of momentums was first noticed in the case $\beta^2 = 1$ by Runkel and Watts [11], and we will prove it for $\beta^2 \in \mathbb{R}_{>0} - \mathbb{Q}$ in Section 3. So we consider limits such that $P_{\langle r,s\rangle} \to P$ for an arbitrary $P \in \mathbb{R}$, which typically implies $r,s \to \infty$.

The limit of the diagonal sector is therefore

$$\lim_{\substack{p,q\to\infty \\ \frac{p}{q}\to\beta^2}} \frac{1}{2} \bigoplus_{\substack{(r,s)\in K_{p,q} \\ rs\in\mathbb{Z}+\frac{1}{2}+\frac{pq}{4}}} \left|\mathcal{R}_{\langle r,s\rangle}\right|^2 = \int_{\mathbb{R}_+} dP \, \left|\mathcal{V}_P\right|^2. \tag{2.14}$$

At this stage, we can only guess the multiplicity of each Verma module $\left|\mathcal{V}_P\right|^2$: in principle, this could be any integer or even infinity. We anticipate that correlation functions only depend on conformal dimensions (2.6), which means that each Verma module should have multiplicity one. Due to the relation $\mathcal{V}_P = \mathcal{V}_{-P}$, the limit of the diagonal sector involves an integral of the type $\int_{\mathbb{R}_+} = \frac{1}{2}\int_{\mathbb{R}}$.

**Fields and fusion rules**

Using the state-field correspondence, let us introduce primary fields that correspond to the highest-weight states in our highest-weight representations. We call $V_P^D$ and $V_{\langle r,s\rangle}^N$ the primary fields that are respectively associated to the representations $|\mathcal{V}_P|^2$ and $\mathcal{V}_{P_{\langle r,s\rangle}} \otimes \bar{\mathcal{V}}_{P_{\langle -r,s\rangle}}$. Operator product expansions of these fields obey algebraic constraints called fusion rules.

In D-series minimal models, there are two types of fusion rules: constraints from the model's $\mathbb{Z}_2$ symmetry [10], which we call conservation of diagonality, and constraints from the fact that fields are degenerate. In our limit, the fields become non-degenerate, which suggests that the second type of fusion rules disappear. The alert reader may raise an objection from the work of Runkel and Watts [11], who studied the limit of diagonal minimal models when $\frac{p}{q} \to 1$ with $q = p+1$, and found non-degenerate fields that obey nontrivial fusion rules reflecting their degenerate origin. However, this survival of the degenerate fusion rules is an artefact of $\frac{p}{q}$ having a rational limit, and of reaching that limit in a specific way. It relies on a delicate mechanism that does not occur in our limit (2.9), and also would not occur in the limit $\frac{p}{q} \to 1$ with say $q = p + O(\sqrt{p})$.

Therefore, the only fusion rule in our limit of D-series minimal models is the conservation of diagonality, and the OPEs are of the type

$$V_{P_1}^D V_{P_2}^D \sim \int_{\mathbb{R}_+} dP \; V_P^D \;, \tag{2.15}$$

$$V_{P_1}^D V_{\langle r_2,s_2\rangle}^N \sim \sum_{r\in 2\mathbb{Z}} \sum_{s\in\mathbb{Z}+\frac{1}{2}} V_{\langle r,s\rangle}^N \;, \tag{2.16}$$

$$V_{\langle r_1,s_1\rangle}^N V_{\langle r_2,s_2\rangle}^N \sim \int_{\mathbb{R}_+} dP \; V_P^D \;. \tag{2.17}$$

We will also need correlation functions that involve diagonal degenerate fields. Let $V_{\langle r_1,s_1\rangle}^D$ be a diagonal degenerate field associated to the representation $\left|\mathcal{R}_{\langle r_1,s_1\rangle}\right|^2$ with $r_1, s_1 \in \mathbb{N}^*$, then its OPE with a diagonal field of momentum $P_2$ is of the type

$$V_{\langle r_1,s_1\rangle}^D V_{P_2}^D \sim \sum_{r=-\frac{r_1-1}{2}}^{\frac{r_1-1}{2}} \sum_{s=-\frac{s_1-1}{2}}^{\frac{s_1-1}{2}} V_{P_2+r\beta+s\beta^{-1}}^D \;, \tag{2.18}$$

where the indices $i, j$ belong to $\frac{1}{2}\mathbb{Z}$ and run by increments of 1, so that the sum has $r_1 s_1$ terms. In a minimal model, fields are doubly degenerate i.e. $V_{\langle r_1,s_1\rangle}^D = V_{\langle q-r_1,p-s_1\rangle}^D$, and their fusion rules are more complicated and depend on the indices $p, q$ (as reviewed in [3]). However, for reasons that we will explain in Section 2.3, we will actually not need doubly degenerate fields and their fusion rules.

## 2.2 Structure constants and their signs: the tricky bit

**Analytic bootstrap techniques**

The analytic bootstrap techniques for determining structure constants rely on the assumption that there exist degenerate fields, and on the crossing symmetry of four-point functions that involve degenerate fields. It is particularly natural to use these techniques in the case of minimal models, whose spectrums are made of degenerate representations [12]. These techniques can also be applied to Liouville theory, whose spectrum is continuous and does not contain any degenerate representation. For this, we need the additional assumption that correlation

functions depend analytically on the fields' momentums [6]. Furthermore, these techniques have recently been generalized to non-diagonal theories [5]. The resulting structure constants are simply related to Liouville theory structure constants, although this relation obscures their analytic properties, and does not determine their signs. We will see that their signs play an essential role in the non-rational limit.

In the non-rational limit of D-series minimal models, there are no degenerate states in the spectrum. Nevertheless, degenerate fields still exist as limits of minimal models' degenerate fields. This provides an a priori justification for the use of degenerate fields. In addition, we will validate the results a posteriori by checking crossing symmetry in four-point functions without degenerate fields. Crossing symmetry in four-point functions with one degenerate field is indeed enough for determining structure constants, but does not imply crossing symmetry in more general four-point functions, see [3] for a discussion in the context of Liouville theory.

In order to use the analytic bootstrap techniques, we also need the analyticity assumption. We will see that some correlation functions obey the analyticity assumption, and can be straightforwardly determined by directly applying the results of [5]. Some other correlation functions violate the analyticity assumption: understanding them is the main goal of this article.

**Two- and three-point structure constants**

Conformal symmetry determines two- and three-point correlation functions of primary fields up to factors called structure constants, which do not depend on the fields' positions. (See [3] for a review.) Thanks to the conservation of diagonality, there are two types of non-vanishing two-point functions:

$$\left\langle V_{P_1}^D V_{P_2}^D \right\rangle = B_{P_1} \delta(P_1 - P_2) \quad , \quad \left\langle V_{\langle r_1, s_1 \rangle}^N V_{\langle r_2, s_2 \rangle}^N \right\rangle = B_{\langle r_1, s_1 \rangle} \delta_{r_1, s_1} \delta_{r_2, s_2} \,, \tag{2.19}$$

where we omit the dependence of the fields and correlation functions on the fields' positions, and introduce the two-point structure constants $B_P$ and $B_{\langle r,s \rangle}$. In rational theories such as minimal models, fields are usually normalized such that two-point structure constants are one. However, we will choose another normalization which makes the analytic properties of correlation functions more manifest. Again thanks to the conservation of diagonality, there are two types of non-vanishing three-point functions:

$$\left\langle V_{P_1}^D V_{P_2}^D V_{P_3}^D \right\rangle = C_{P_1, P_2, P_3} \quad , \quad \left\langle V_{P_1}^D V_{\langle r_2, s_2 \rangle}^N V_{\langle r_3, s_3 \rangle}^N \right\rangle = C_{P_1, \langle r_2, s_2 \rangle, \langle r_3, s_3 \rangle} \,. \tag{2.20}$$

Let us reproduce the results of [5] for the structure constants. (Our normalizations correspond to $Y = 1$ in that work.) The two-point structure constants are

$$B_P = \prod_\pm \Upsilon_\beta(\beta \pm 2P) \,, \tag{2.21}$$

$$B_{\langle r,s \rangle} = \frac{(-1)^{rs}}{\prod_\pm \Gamma_\beta(\beta \pm 2P_{\langle r,s \rangle}) \Gamma_\beta(\beta^{-1} \pm 2P_{\langle -r,s \rangle})} \,, \tag{2.22}$$

where $\Gamma_\beta$ is Barnes' double Gamma function, and $\Upsilon_\beta(x) = \frac{1}{\Gamma_\beta(x)\Gamma_\beta(\beta + \beta^{-1} - x)}$ is the Upsilon function. The diagonal three-point structure constant is the same as in Liouville theory,

$$C_{P_1, P_2, P_3} = \prod_{\pm, \pm} \Upsilon_\beta \left( \tfrac{\beta + \beta^{-1}}{2} + P_1 \pm P_2 \pm P_3 \right) \,. \tag{2.23}$$

The Upsilon function is analytic on the complex plane, with

$$\Upsilon_\beta \left( \tfrac{\beta + \beta^{-1}}{2} + x \right) = 0 \iff x \in \pm \left( \beta(\mathbb{N} + \tfrac{1}{2}) + \beta^{-1}(\mathbb{N} + \tfrac{1}{2}) \right) \,. \tag{2.24}$$

The resulting zeros of the diagonal three-point structure constant will lead to simplifications in the four-point functions of Section 4.2. Finally, the non-diagonal three-point structure constant is

$$C_{P_1,\langle r_2,s_2\rangle,\langle r_3,s_3\rangle} =$$
$$\frac{(-1)^{r_2 s_3}\sigma(P_1)}{\prod_{\pm,\pm}\Gamma_\beta(\frac{\beta+\beta^{-1}}{2}+P_1\pm P_{\langle r_2,s_2\rangle}\pm P_{\langle r_3,s_3\rangle})\prod_{\pm,\pm}\Gamma_\beta(\frac{\beta+\beta^{-1}}{2}-P_1\pm P_{\langle -r_2,s_2\rangle}\pm P_{\langle -r_3,s_3\rangle})} \ . \quad (2.25)$$

For non-diagonal fields that belong to D-series minimal models or to their limit (2.11), our assumption (2.10) implies $r_i \in 2\mathbb{Z}$ and $s_i \in \mathbb{Z}+\frac{1}{2}$. Let us briefly discuss the two sign factors $(-1)^{r_2 s_3}$ and $\sigma(P_1)$. First, the sign $(-1)^{r_2 s_3}$ is here to ensure that the structure constant has the correct behaviour under permuting the non-diagonal fields, $C_{P_1,\langle r_2,s_2\rangle,\langle r_3,s_3\rangle} = (-1)^{r_2 s_2 + r_3 s_3} C_{P_1,\langle r_3,s_3\rangle,\langle r_2,s_2\rangle}$ [3]. Second, the sign factor $\sigma(P)$ is defined by the shift equations

$$\sigma(P+\beta^{-1}) = \sigma(P) \quad , \quad \sigma(P+\beta) = -\sigma(P) \ . \quad (2.26)$$

(This sign factor was called $f_{2,3}(P_1)$ in [5](3.39), and it depends on the parities of the integers $2r_2, 2r_3, 2s_2, 2s_3$; in our case $2r_i$ is even and $2s_i$ odd, hence the signs + and − in our shift equations.)

Notice that the two fields $V^N_{\langle 0,s\rangle}$ and $V^D_{P_{\langle 0,s\rangle}}$ have the same left and right momentums $(P,\bar{P}) = (P_{\langle 0,s\rangle}, P_{\langle 0,s\rangle})$. These two fields however do not coincide, and in particular their three-point structure constants differ by sign factors,

$$C_{P_1,\langle 0,s_2\rangle,\langle 0,s_3\rangle} = \sigma(P_1)C_{P_1,P_{\langle 0,s_2\rangle},P_{\langle 0,s_3\rangle}} \ . \quad (2.27)$$

**The sign problem**

The existence of solutions of the shift equations (2.26) depends on the allowed values of the momentum $P$. Let us begin with the case of a non-diagonal minimal model. Given our assumptions (2.10) on the parities of $p$ and $q$, the diagonal sector of the spectrum $\mathcal{S}^{\text{D-series}}_{p,q}$ (2.7) is made of representations whose indices belong to the finite set

$$(r,s) \in \left[\left(2\mathbb{Z}+\tfrac{1}{2}+\tfrac{pq}{2}\right)\cap\left(-\tfrac{q}{2},\tfrac{q}{2}\right)\right] \times \left[\left(\mathbb{Z}+\tfrac{1}{2}\right)\cap\left(-\tfrac{p}{2},\tfrac{p}{2}\right)\right] . \quad (2.28)$$

The corresponding momentums $P_{\langle r,s\rangle}$ differ by elements of $\beta\mathbb{Z}+\frac{\beta^{-1}}{2}\mathbb{Z}$. Given the additional requirement $\sigma(P) = \sigma(-P)$, the shift equations have a unique solution on this finite set, up to a constant prefactor which we set to one:

$$\sigma(P_{\langle r,s\rangle}) = (-1)^{\frac{r}{2}} \quad , \quad \text{(D-series minimal model)} \ . \quad (2.29)$$

Similarly, let us consider the shift equations for momentums that result from the fusion of a degenerate field (2.18). These momentums form a finite set whose elements differ by elements of $\beta\mathbb{Z}+\beta^{-1}\mathbb{Z}$. The shift equations again have a unique solution on this set:

$$\sigma(P_2+r\beta+s\beta^{-1}) = (-1)^r \quad , \quad \text{(fusion product of a degenerate field)} \ . \quad (2.30)$$

(The difference between these sign factors $(-1)^{\frac{r}{2}}$ and $(-1)^r$ is due to different conventions for the index $r$, in particular $P_{\langle r+1,s\rangle} = P_{\langle r,s\rangle}+\frac{1}{2}\beta$.)

Finally, let us consider the set $P \in \mathbb{R}_+$ of allowed momentums in the limit (2.14) of the diagonal spectrum of the minimal models, while assuming $\beta^2 \in \mathbb{R}_+ - \mathbb{Q}$. If the second shift

equation was $\sigma(P + \beta) = \sigma(P)$, the constant function $\sigma(P) = 1$ would be the unique continuous solution of the shift equations, by the very argument that leads to the uniqueness of the three-point function in Liouville theory [6]. However, we do have a minus sign in the second shift equation, so the shift equations have no continuous solution $\sigma(P)$.

This shows that in the non-rational limit of D-series minimal models, correlation functions cannot be analytic (or even continuous) as functions of the momentum $P$ of the diagonal sector. The analytic bootstrap techniques are not directly applicable, and we will have to take the limit of minimal models' correlation functions.

## 2.3 The fixed $c$ limit

**From the limit of minimal models to the limit of degenerate fields**

Taking the limit of minimal models' correlation functions is a messy business, as it involves the dependence of correlation functions on both the central charge and the fields' momentums. However, the only reason why we need minimal models is for approximating non-analytic expressions in the limit theory. Let us look for simpler approximations that are still free of the sign problem.

The sign problem occurs when momentums belong to a continuum, and is absent when momentums belong to finite sets. In particular, the problem is absent from minimal models, whose spectrums are finite. But we do not really need minimal models for making the spectrum finite: for any value of the central charge, any four-point function that involves a degenerate field has a finite spectrum, in the sense that the sum over representations in the decomposition (2.2) is finite by virtue of the degenerate fields' fusion rules (2.18).

Therefore, in any four-point function with at least one diagonal field, we can solve the sign problem by approximating that field with diagonal degenerate fields, without changing the central charge or the other three fields,

$$V_{P_1}^D = \lim_{\substack{r_1,s_1 \in \mathbb{N}^* \\ r_1,s_1 \to \infty \\ \beta r_1 - \beta^{-1} s_1 \to 2P_1}} V_{\langle r_1,s_1 \rangle}^D . \tag{2.31}$$

This fixed $c$ limit should agree with the limit from minimal models, because our correlation functions would be analytic functions of the momentums and of the central charge, if there was no sign problem.

**Towards a mathematical formulation**

Let us consider a fixed irrational value of the central charge i.e. $\beta^2 \in \mathbb{R}_+ - \mathbb{Q}$. Consider a four-point function in the decomposition (2.2), assuming that the first two fields $V_{\langle r_1,s_1 \rangle}^D$ and $V_{P_2}^D$ are diagonal, the first one being degenerate with $r_1, s_1 \in \mathbb{N}^*$:

$$
\begin{array}{cc}
\begin{array}{c}
P_2 \diagdown \qquad \diagup N \\
\overline{P_2 + r\beta + s\beta^{-1}} \\
\langle r_1, s_1 \rangle \diagup \qquad \diagdown N \\
\Phi^-
\end{array}
&
\begin{array}{c}
P_2 \diagdown \qquad \diagup D \\
\overline{P_2 + r\beta + s\beta^{-1}} \\
\langle r_1, s_1 \rangle \diagup \qquad \diagdown D \\
\Phi^+
\end{array}
\end{array}
\tag{2.32}
$$

We will shortly write the four-point function as a finite sum $\Phi^{\pm}$ according to the fusion rules (2.18). In the case of $\Phi^-$, the summand involves the sign factor $(-1)^r$ (2.30) from one of the

three-point structure constants, so we have an alternating sum. Our four-point function is of the type

$$\Phi^\pm_{(r_1,s_1)}[f] = \sum_{r=-\frac{r_1-1}{2}}^{\frac{r_1-1}{2}} \sum_{s=-\frac{s_1-1}{2}}^{\frac{s_1-1}{2}} (\pm 1)^r f\left(P_2 + r\beta + s\beta^{-1}\right), \tag{2.33}$$

where the analytic function $f$ is the summand of the decomposition (2.2), after omitting the possible sign factor. We want to find the limit of this expression when the degenerate field $V^D_{\langle r_1,s_1\rangle}$ tends to a non-degenerate field with an arbitrary momentum $P_1 \in \mathbb{R}$, i.e.

$$\Phi^\pm_{P_1}[f] = \lim_{\substack{r_1,s_1\in\mathbb{N}^* \\ r_1,s_1\to\infty \\ \beta r_1-\beta^{-1}s_1\to 2P_1}} \Phi^\pm_{(r_1,s_1)}[f]. \tag{2.34}$$

Actually, our function $f$ depends on $r_1, s_1$ via the momentum $P_{\langle r_1,s_1\rangle}$. This dependence is however analytic, and we can treat $f$ as independent from $r_1, s_1$. Moreover, due to its conformal block factor, the function $f$ has poles at certain real values of the momentum. However, we can avoid all these poles by assuming that $P_2$ is not real. Then we only need evaluate conformal blocks on the line $\mathbb{R}+i\Im P_2$, where they are analytic, and have a Gaussian-like decrease at infinity. Analyticity and Gaussian-like decrease actually hold on any strip of the type $\{\eta < \Im P < M\}$ for $\eta > 0$.

## 3 Sums over squashed lattices

In this mathematical interlude, we compute the limit $\Phi^\pm_{P_1}[f]$ (2.34) for a function $f(P)$ that is analytic and has a Gaussian-like decrease at infinity on strips of the type $\{\eta < \Im P < M\}$. These conditions on $f$ are certainly stronger than needed, but they are fulfilled in our CFT problem, so we do no try to weaken them. We also make the technical assumption $r_1 \in 4\mathbb{N} + 1$, which will spare us a few sign factors, without otherwise changing the results.

At first sight, the limit may seem to be given by Eq. (2.33) with $r_1, s_1 = +\infty$, i.e. by a sum over $P_2 + \beta\mathbb{Z} + \beta^{-1}\mathbb{Z}$. We call this set a squashed lattice because it would be a two-dimensional lattice if $\beta^2 \notin \mathbb{R}$, and reduces to a subset of a line for $\beta^2 \in \mathbb{R}$. But in a sum over a squashed lattice, the argument $P_2 + r\beta + s\beta^{-1}$ of the function $f$ does not go to infinity when say $r \to \infty, s \to -\infty$. So there is no reason for $\Phi^\pm_{(r_1,s_1)}[f]$ to have a well-defined limit for $r_1, s_1 \to \infty$, and it is essential that we impose the additional condition $\beta r_1 - \beta^{-1}s_1 \to 2P_1$.

**Performing the first one of the two sums**

Our first step is to make the sum over $s$ infinite, which makes sense so long the sum over $r$ remains finite. We use the identity

$$\sum_{s=-\frac{s_1-1}{2}}^{\frac{s_1-1}{2}} \varphi(s) = \sum_{s\in\mathbb{Z}+\frac{1}{2}} \varphi(s+\tfrac{s_1}{2}) - \sum_{\epsilon=\pm} \sum_{s\in\mathbb{N}+\frac{1}{2}} \varphi\left(\epsilon(s+\tfrac{s_1}{2})\right), \tag{3.1}$$

for any function $\varphi$ such that the sums converge. We also use the fact that in our limit $\beta^{-1}s_1 \sim -2P_1 + \beta r_1$. We obtain

$$
\Phi_{P_1}^{\pm}[f] = \lim_{\substack{r_1,s_1 \in \mathbb{N}^* \\ r_1,s_1 \to \infty \\ \beta r_1 - \beta^{-1}s_1 \to 2P_1}} \sum_{r=\frac{1}{2}}^{r_1-\frac{1}{2}} \sum_{s \in \mathbb{Z}+\frac{1}{2}} (\pm 1)^{r-\frac{1}{2}} f(P_2 - P_1 + \beta r + \beta^{-1}s)
$$
$$
- \sum_{r,s \in \mathbb{N}+\frac{1}{2}} \sum_{\epsilon=\pm} (\pm 1)^{r-\frac{1}{2}} f\left(P_2 + \epsilon(\beta r + \beta^{-1}s - P_1)\right) , \quad (3.2)
$$

where we have shifted the indices $r,s$, using our technical assumption $r_1 \in 4\mathbb{N}+1$ to deal with the sign prefactor. We performed the limit in the second term thanks to $\lim_{r,s \to +\infty}(\beta r + \beta^{-1}s) = \infty$.

It remains to perform the limit in the first term. Since the sum over $s$ is infinite, the sum over $r$ adds values of a periodic function, with the period $\beta^{-1}$. We Fourier transform that periodic function, using the identity

$$
\sum_{s \in \mathbb{Z}+\frac{1}{2}} f\left(P_0 + \beta^{-1}s\right) = \beta \sum_{n \in \mathbb{Z}} (-1)^n e^{2\pi i \beta n P_0} \int_{\mathbb{R}+P_0} f(P) e^{-2\pi i \beta n P} dP , \quad (3.3)
$$

where the function $f$ is assumed to be analytic on $\mathbb{R} + P_0$. After this Fourier transformation, the sum over $r$ in Eq. (3.2) is geometric.

**Alternating sums**

Let us first perform the alternating geometric sum

$$
\lim_{\substack{r_1,s_1 \in \mathbb{N}^* \\ r_1,s_1 \to \infty \\ \beta r_1 - \beta^{-1}s_1 \to 2P_1}} \sum_{r=\frac{1}{2}}^{r_1-\frac{1}{2}} (-1)^{r-\frac{1}{2}} e^{2\pi i \beta n (P_2-P_1+\beta r)} = e^{2\pi i n \beta P_2} \frac{\cos(2\pi n \beta P_1)}{\cos(\pi n \beta^2)} , \quad (3.4)
$$

which leads to the result

$$
\Phi_{P_1}^{-}[f] = \beta \sum_{n \in \mathbb{Z}} (-1)^n \frac{\cos(2\pi n \beta P_1)}{\cos(\pi n \beta^2)} \int_{\mathbb{R}} f(P + P_2) \cos(2\pi n \beta P) dP
$$
$$
- \sum_{r,s \in \mathbb{N}+\frac{1}{2}} \sum_{\epsilon=\pm} (-1)^{r-\frac{1}{2}} f\left(P_2 + \epsilon(\beta r + \beta^{-1}s - P_1)\right) . \quad (3.5)
$$

We may be tempted to exchange the sum over $n$ with the integral over $P$, in order to write the first term of $\Phi_{P_1}^{-}[f]$ as an integral of $f$ against some density. The expression for that density would however be a divergent sum over $n$, namely

$$
\omega(P) = \beta \sum_{n \in \mathbb{Z}} (-1)^n \frac{\cos(2\pi n \beta P_1) \cos(2\pi n \beta P)}{\cos(\pi n \beta^2)} . \quad (3.6)
$$

This is not a function, but a more general distribution, just like the Dirac delta function. However, this is also not a linear combination of Dirac delta functions, except in special cases such as $P_1 = \frac{\beta}{2}$. Intuitively, this is because our squashed lattice is dense in the real $P$-line, so the support of our distribution should be $\mathbb{R}$ itself. For more properties of distributions of this type, see Section 4.3.

**Convergence of alternating sums**

Under our assumptions on $f$, let us discuss the convergence of the sums and integral in $\Phi^-_{P_1}[f]$ (3.5). Obviously the double sum over $r, s$ converges, and the integral over $P$ converges too. We are left with discussing the convergence of the sum over $n$. Loosely speaking, since $f(P + \mathfrak{J}P_2)$ is analytic on a strip of width $|\mathfrak{J}P_2|$, its Fourier transform decreases like $e^{-2\pi|\beta\mathfrak{J}P_2 n|}$ as $n \to \infty$. This statement (or rather its more precise version) implies that the sum over $n$ converges under the two conditions

- $|\mathfrak{J}P_1| < |\mathfrak{J}P_2|$,

- $\frac{1}{\cos(\pi n \beta^2)}$ grows less than exponentially.

We thus need to bound $|\cos(\pi n \beta^2)|$ from below. This quantity can vanish only if $\beta^2$ is rational. How small it can get as $n \to \infty$ depends on how well $\beta^2$ can be approximated by rational numbers, in other words on the Diophantine approximations of $\beta^2$. We now assume that $\beta^2$ is not a Liouville number, which means

$$\exists m, q_0 \in \mathbb{N} \ , \ \ \forall (p, q) \in \mathbb{Z} \times \mathbb{Z}_{>q_0} \ , \ \ \left| \beta^2 - \frac{p}{q} \right| > \frac{1}{q^m} \ . \tag{3.7}$$

This implies that $\frac{1}{\cos(\pi n \beta^2)}$ is polynomially bounded as $n \to \infty$, which is enough for our sum over $n$ to converge. Now the set of Liouville numbers is of measure zero in the real line. Therefore, we do not lose much by excluding them, and the alternating sums converge for generic values of $\beta^2$.

While Liouville numbers help us prove convergence for generic $\beta^2$, they are not expected to play any special role in conformal field theory. Depending on the function $f$, the alternating sum may converge for some or most Liouville numbers. In this respect, the properties of the particular functions $f$ that appear in conformal field theory are not obvious.

**Non-alternating sums**

Similarly, let us perform the non-alternating geometric sum

$$\lim_{\substack{r_1, s_1 \in \mathbb{N}^* \\ r_1, s_1 \to \infty \\ \beta r_1 - \beta^{-1} s_1 \to 2P_1}} \sum_{r = \frac{1}{2}}^{r_1 - \frac{1}{2}} e^{2\pi i \beta n (P_2 - P_1 + \beta r)} = e^{2\pi i n \beta P_2} \frac{\sin(2\pi n \beta P_1)}{\sin(\pi n \beta^2)} \ . \tag{3.8}$$

This expression is however valid for $n \neq 0$ only. The result for $n = 0$ cannot be deduced from this expression, as the limits $r_1, s_1 \to \infty$ and $n \to 0$ do not commute. Rather, the result for $n = 0$ is simply

$$\lim_{r_1 \to \infty} \sum_{r = \frac{1}{2}}^{r_1 - \frac{1}{2}} 1 = \lim_{r_1 \to \infty} r_1 = \infty \ . \tag{3.9}$$

The presence of the infinite $n = 0$ term allows us to neglect the finite $n \neq 0$ terms, and also the discrete terms in the second line of Eq. (3.2), and we find

$$\Phi^+_{P_1}[f] = \infty \times \int_{\mathbb{R} + P_2} f(P) dP \ . \tag{3.10}$$

There are simpler ways to compute $\Phi^+_{P_1}[f]$. Going back to the original expression (2.33) of $\Phi^+_{(r_1, s_1)}[f]$, we see that $\Phi^+_{P_1}[f]$ must come with an infinite prefactor, because the momentums

$\beta r + \beta^{-1}s$ visit any given real interval an infinite number of times as $r_1, s_1 \to \infty$. Moreover, we may argue that the momentums' distribution becomes invariant under shifts by $\beta$ and $\beta^{-1}$, and therefore uniform, directly leading to the result (3.10). We take this argument as a sanity check of the manipulations that we used to compute $\Phi_{P_1}^+[f]$ and $\Phi_{P_1}^-[f]$.

# 4 Four-point functions and crossing symmetry

Let us use the results of Section 3 for computing the limits of four-point functions of D-series minimal models, when written in their conformal block decompositions. Due to the conservation of diagonality, a correlation function can be nonvanishing only if it involves an even number of non-diagonal fields. We therefore have three types of nonvanishing four-point functions, with 0, 2 or 4 non-diagonal fields, which we respectively called diagonal, mixed and non-diagonal. We will start with the diagonal and non-diagonal case, which are simple because they do not involve the sign problem of Section 2.2. We will then deal with the harder and more interesting mixed case.

## 4.1 Diagonal and non-diagonal four-point functions

### Analytic bootstrap

Let us forget our limit of D-series minimal models for a moment, and go back to the analytic bootstrap as reviewed in Section 2.

Diagonal four-point functions do not have a sign problem, because their decompositions into conformal blocks only involve diagonal three-point structure constants. They are actually identical to four-point functions of Liouville theory with $c \leq 1$, which are uniquely determined by the analytic bootstrap equations for diagonal theories.

Non-diagonal four-point functions also do not have a sign problem, because the signs from the two non-diagonal structure constants cancel. Explicitly, the decomposition (2.2) reads

$$\left\langle \prod_{i=1}^{4} V_{\langle r_i, s_i \rangle}^N(z_i) \right\rangle = \int dP \frac{C_{P, \langle r_1, s_1 \rangle, \langle r_2, s_2 \rangle} C_{P, \langle r_3, s_3 \rangle, \langle r_4, s_4 \rangle}}{B_P} \mathcal{F}_P^{(s)}(\{z_i\}), \qquad (4.1)$$

where the product of the two non-diagonal three-point structure constants (2.25) involves the squared sign factor $\sigma(P)^2$. The solution of the shift equations (2.26) for $\sigma(P)^2$ is simply $\sigma(P)^2 = 1$, and the decomposition's integrand depends analytically on $P$.

In both cases, the integration line $P \in \mathbb{R}$ encounters poles of the conformal blocks. This problem was solved in the context of Liouville theory, by shifting the integration line to $\mathbb{R} + i\epsilon$ with $\epsilon \in \mathbb{R}^*$. Then the integral converges, and does not depend on $\epsilon$ [13].

Notice that diagonal and non-diagonal four-point functions sometimes coincide. Due to the relation (2.27) between three-point structure constants, and the cancellation of sign factors, we indeed have

$$\left\langle \prod_{i=1}^{4} V_{\langle 0, s_i \rangle}^N \right\rangle = \left\langle \prod_{i=1}^{4} V_{P_{\langle 0, s_i \rangle}}^D \right\rangle. \qquad (4.2)$$

This coincidence will allow us to deduce some properties of non-diagonal four-point functions from the well-known properties of the Liouville theory four-point functions.

### The limit from minimal models and its divergence

Our original motivation for taking the limit from minimal models is the failure of the analytic bootstrap in the presence of the sign problem. We are now dealing with four-point functions

that have no sign problem: let us nevertheless discuss the limit, for the sake of understanding its properties.

For diagonal four-point functions, our fixed $c$ limit of non-alternating sums (3.10) agrees with the analytic bootstrap result, including the shift of the integration line to complex momentums. The infinite prefactor in the fixed $c$ limit only means that we should add a prefactor to the limit (2.31) in order to make it finite, namely

$$\left\langle \prod_{i=1}^{4} V_{P_i}^D \right\rangle = \lim_{\substack{r_1, s_1 \in \mathbb{N}^* \\ r_1, s_1 \to \infty \\ \beta r_1 - \beta^{-1} s_1 \to 2P_1}} \frac{1}{r_1} \left\langle V_{\langle r_1, s_1 \rangle}^D V_{P_2}^D V_{P_3}^D V_{P_4}^D \right\rangle . \tag{4.3}$$

We would encounter the same divergence if we directly considered a limit from D-series minimal models, rather than our technically simpler fixed $c$ limit. The divergence is due to the sum over $r$ in the degenerate fusion rule (2.18) becoming infinite; in the limit (2.9) of the minimal models $\mathrm{MM}_{p,q}$ the bound on $r$ would be of order $q$ (or equivalently $p$), and we would find

$$\left\langle \prod_{i=1}^{4} V_{P_i}^D \right\rangle = \lim_{\substack{p, q \to \infty \\ \frac{p}{q} \to \beta^2 \\ \beta r_i - \beta^{-1} s_i \to 2P_i}} \frac{1}{q} \left\langle \prod_{i=1}^{4} V_{\langle r_i, s_i \rangle}^D \right\rangle_{\mathrm{MM}_{p,q}} . \tag{4.4}$$

(This may actually differ from Eq. (4.3) by a finite factor that depends solely on $\beta$.)

For non-diagonal four-point functions, our fixed $c$ limit does not make sense, since non-diagonal fields have discrete momentums, and cannot be approximated by degenerate fields. However, we can still take limits of minimal models. Based on the agreement (4.2) with diagonal four-point functions in special cases, we expect that limit to behave in the same way as the limit of diagonal four-point functions. In particular, we need a prefactor $\frac{1}{q}$ for making the limit finite,

$$\left\langle \prod_{i=1}^{4} V_{\langle r_i, s_i \rangle}^N \right\rangle = \lim_{\substack{p, q \to \infty \\ \frac{p}{q} \to \beta^2}} \frac{1}{q} \left\langle \prod_{i=1}^{4} V_{\langle r_i, s_i \rangle}^N \right\rangle_{\mathrm{MM}_{p,q}} , \tag{4.5}$$

where the indices $r_i, s_i$ are fixed, and we assume $p, q$ to be large enough for the corresponding representations to belong to the Kac table.

**Numerical checks of crossing symmetry**

Diagonal four-point functions belong to Liouville theory with $c \leq 1$, and their crossing symmetry was already checked in [13]. We therefore focus on non-diagonal four-point function. Our results lead to the prediction of a large class of crossing-symmetric four-point functions, depending on the continuous parameter $\beta^2 \in \mathbb{R}_{>0}$, and on four pairs of discrete indices $(r_i, s_i) \in 2\mathbb{Z} \times (\mathbb{Z} + \frac{1}{2})$. Any given four-point function moreover depends on the cross-ratio $z \in \mathbb{C}$ of the four fields' positions.

We numerically find that crossing symmetry is obeyed to a good accuracy, and that discrepancies can be attributed to the approximations that we use in the numerical calculations: the truncations of sums and integrals, and the finite depth in the computation of conformal blocks using Zamolodchikov's recursion. For ease of graphical representation, we focus on the segment $z = x + 0.4i$ with $x \in (-0.5, 1.5)$. On this segment, we computed the four-point function $\left\langle V_{\langle 0, \frac{3}{2} \rangle}^N V_{\langle 4, \frac{1}{2} \rangle}^N V_{\langle 2, \frac{5}{2} \rangle}^N V_{\langle 2, -\frac{1}{2} \rangle}^N \right\rangle$ at $c = -0.41$. We found an excellent agreement between the three channels, with $5 - 10$ common digits for most values of $x$. Here is a plot of the real and imaginary parts of this four-point function:

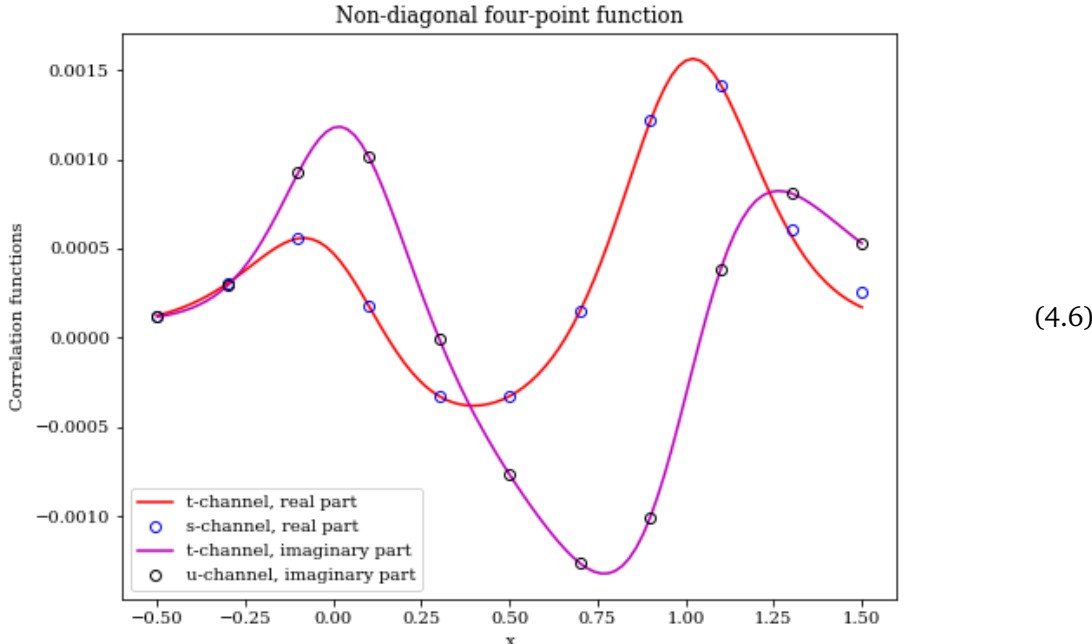

(4.6)

## 4.2 Mixed four-point functions

**Convergence of the limits**

We consider a four-point function of the type $\left\langle V_{P_1}^D V_{P_2}^D V_{\langle r_3,s_3\rangle}^N V_{\langle r_4,s_4\rangle}^N \right\rangle$. By conservation of diagonality, we expect a diagonal spectrum in the $s$-channel decomposition, and a non-diagonal spectrum in the $t$- and $u$-channel decompositions:

$$
\begin{matrix} D & & N \\ & D & \\ D & & N \end{matrix} \quad = \quad \begin{matrix} D & & N \\ & N & \\ D & & N \end{matrix} \quad = \quad \begin{matrix} D & & N \\ & N & \\ D & & N \end{matrix} \tag{4.7}
$$

While this article is about understanding the $s$-channel decomposition, the other two channels have already been studied in [5], where their agreement was checked. Moreover, the fixed $c$ limit and the limit from minimal models are unproblematic in these channels: approximating one or more fields with degenerate fields amounts to truncating the non-diagonal spectrum (2.11) to a finite subset, and taking the limit removes the truncation. Therefore,

$$
\left\langle V_{P_1}^D V_{P_2}^D V_{\langle r_3,s_3\rangle}^N V_{\langle r_4,s_4\rangle}^N \right\rangle = \lim_{\substack{r_1,s_1\in\mathbb{N}^* \\ r_1,s_1\to\infty \\ \beta r_1 - \beta^{-1}s_1 \to 2P_1}} \left\langle V_{\langle r_1,s_1\rangle}^D V_{P_2}^D V_{\langle r_3,s_3\rangle}^N V_{\langle r_4,s_4\rangle}^N \right\rangle, \tag{4.8}
$$

$$
= \lim_{\substack{p,q\to\infty \\ \frac{p}{q}\to\beta^2 \\ \beta r_1 - \beta^{-1}s_1 \to 2P_1 \\ \beta r_2 - \beta^{-1}s_2 \to 2P_2}} \left\langle V_{\langle r_1,s_1\rangle}^D V_{\langle r_2,s_2\rangle}^D V_{\langle r_3,s_3\rangle}^N V_{\langle r_4,s_4\rangle}^N \right\rangle_{\mathrm{MM}_{p,q}}. \tag{4.9}
$$

It follows that the $s$-channel decomposition should also have a finite fixed $c$ limit and a finite limit from minimal models.

**Simplification and symmetrization**

Let us evaluate the limit $\Phi^-_{P_1}[f]$ (3.5) for a test function $f$ that is the integrand of the $s$-channel decomposition of our mixed four-point function, after omitting the sign factor $\sigma(P)$:

$$f(P) = \frac{C_{P,P_1,P_2} C_{P,\langle r_3,s_3\rangle,\langle r_4,s_4\rangle}}{\sigma(P)B_P} \mathcal{F}^{(s)}_P(\{z_i\}) . \tag{4.10}$$

This function depends analytically on $P$, except for poles on the real $P$-line, and we can apply our result for $\Phi^-_{P_1}[f]$.

We first notice an important simplification: the discrete momentums $P_2 + \epsilon(\beta r + \beta^{-1}s - P_1)$ that appear on the second line of the limit $\Phi^-_{P_1}[f]$ (3.5) fall on zeros of the diagonal three-point structure constant $C_{P,P_1,P_2}$ (2.23)-(2.24). Therefore, the corresponding terms of $\Phi^-_{P_1}[f]$ vanish. We are left with the distributional first line,

$$\left\langle V^D_{P_1} V^D_{P_2} V^N_{\langle r_3,s_3\rangle} V^N_{\langle r_4,s_4\rangle} \right\rangle = \beta \sum_{n\in\mathbb{Z}} (-1)^n \frac{\cos(2\pi n\beta P_1)}{\cos(\pi n\beta^2)} \int_{\mathbb{R}} f(P+P_2)\cos(2\pi n\beta P)dP . \tag{4.11}$$

From its $t$- and $u$-channel decompositions, we expect that the four-point function is invariant under $P_1 \leftrightarrow P_2$, and analytic in $P_2$. In order to see whether our $s$-channel decomposition obeys these properties, let us use the parity $f(P) = f(-P)$, which leads to the identity

$$\int_{\mathbb{R}} f(P+P_2)\cos(\nu P)dP$$
$$= \cos(\nu P_2)\int_{\mathbb{R}+i\epsilon} f(P)\cos(\nu P)dP + \sin(\nu P_2)\int_{\mathbb{R}+P_2} f(P)\sin(\nu P)dP , \tag{4.12}$$

where we introduce the temporary notation $\nu = 2\pi\beta n$, and $\epsilon \neq 0$. The first term has the desired invariance and analyticity properties, while the second term does not, in particular it appears to switch sign when $P_2$ crosses the real line. It is tempting to conclude a priori that the second term cannot contribute to the four-point function, but this is not obvious because the split into two terms apparently spoils the convergence of the sum over $n$. A more robust argument comes from the fact that our four-point function is real if $P_1, P_2$ and the fields' positions are real, whereas the second term is purely imaginary in that case,

$$\int_{\mathbb{R}+P_2} f(P)\sin(\nu P)dP = \pi i\,\mathrm{sign}(\mathfrak{I}P_2) \sum_{a\in\mathrm{Poles}(f)} \mathrm{Res}_a(f)\sin(\nu a) . \tag{4.13}$$

Actually, we can further the computation and see that the contribution of any given pole of $f$ to the four-point function vanishes. Our conformal blocks have poles for $P = P_{\langle r,s\rangle}$ with $(r,s) \in 2\mathbb{N}^* \times \mathbb{N}^*$, the simplest case is $P = P_{\langle 2,1\rangle}$. The contribution of this pole is

$$2\pi i\,\mathrm{sign}(\mathfrak{I}P_2)\beta \sum_{n\in\mathbb{Z}} \cos(2\pi n\beta P_1)\sin(2\pi n\beta P_2)\sin(\pi n\beta^2)\mathrm{Res}_{P_{\langle 2,1\rangle}}(f) . \tag{4.14}$$

Assuming $P_1, P_2 \in \mathbb{R}$, this is a combination of Dirac delta functions of the type

$$\sum_{s\in\mathbb{Z}} \delta\left(P_1 + P_2 + \tfrac{\beta}{2} + s\beta^{-1}\right)\mathrm{Res}_{P_{\langle 2,1\rangle}}(f) , \tag{4.15}$$

where we used the identity $\sum_{n\in\mathbb{Z}} e^{2\pi inx} = \sum_{\ell\in\mathbb{Z}} \delta(x+\ell)$ if $x \in \mathbb{R}$. It turns out that $\mathrm{Res}_{P_{\langle 2,1\rangle}}(f)$ vanishes for $P_1 + P_2 + \tfrac{\beta}{2} + s\beta^{-1} = 0$, due to zeros of the structure constant $C_{P_1,P_2,P_{\langle 2,1\rangle}}$ (2.23) if

$s \neq 0$, and to a zero of the conformal block's residue if $s = 0$. Therefore, the pole at $P = P_{\langle 2,1 \rangle}$ does not contribute to the four-point function. By a similar mechanism, we expect that the other poles do not contribute either. While not a paragon of rigour, this argument explains why we can drop the second term in Eq. (4.12): this term is nonzero due to the poles of $f$, but after summing over $n$ it is killed by the zeros of $f$. The result is an expression that is manifestly invariant under $P_1 \leftrightarrow P_2$ and analytic in $P_2$,

$$
\begin{aligned}
&\left\langle V_{P_1}^D V_{P_2}^D V_{\langle r_3, s_3 \rangle}^N V_{\langle r_4, s_4 \rangle}^N \right\rangle \\
&\qquad = \beta \sum_{n \in \mathbb{Z}} (-1)^n \frac{\cos(2\pi n \beta P_1) \cos(2\pi n \beta P_2)}{\cos(\pi n \beta^2)} \int_{\mathbb{R} + i\epsilon} f(P) \cos(2\pi n \beta P) dP .
\end{aligned} \quad (4.16)
$$

However, this expression is not suitable for numerical calculations. This is firstly because the sum over $n$ does not converge fast. And secondly, the integral over $P$ has two sources of instability:

- the poles of $f(P)$ on the real $P$-line,

- the exponential divergence of the cosine factor when its argument is complex.

We need $\epsilon$ to be large for evading the poles, and small for limiting the exponential divergence: it is hard to find values that lead to good numerical precision.

In our original expression (4.11), the cosine factor of the integrand was purely oscillatory, and we could get away from the poles of $f$ by giving $P_2$ a large imaginary part. And the sum over $n$ converged exponentially provided $|\Im P_1| < |\Im P_2|$. This restriction to a region of the $(P_1, P_2)$ space is not a big problem, as we can check crossing symmetry in this region and deduce it elsewhere by analyticity.

**Numerical checks of crossing symmetry**

From Section 4.3, we already know that the $s$-channel decomposition (4.16) of our mixed four-point function converges. Let us now discuss how fast it converges. We will focus on the term that dominates the large $P$ behaviour of the integrand $f(P)$ (4.10),

$$
f(P) \underset{P \to \infty}{\sim} |q|^{2P^2} , \quad (4.17)
$$

where $|q| < 1$ is the nome that corresponds to the cross-ratio of the four fields' positions. (See [3] for a review.) The integral over $P$ converges thanks to $|q| < 1$, and the convergence gets better as $q \to 0$ i.e. $z_1 \to z_2$. So far this is standard behaviour for $s$-channel decompositions of four-point functions. In our case, we still have the sum over $n$ to perform. Our term's large $n$ behaviour is

$$
\int_{\mathbb{R} + i\epsilon} |q|^{2P^2} \cos(2\pi n \beta P) dP \underset{n \to \infty}{\sim} e^{\frac{\pi^2 \beta^2}{\log |q|^2} n^2} . \quad (4.18)
$$

Therefore, the sum over $n$ converges faster for $|q| \to 1$ and slower for $q \to 0$. This suggests that the convergence of the $s$-channel decomposition does not become arbitrarily fast near any particular value of $q$.

Due to these bad convergence properties, and to the restrictions on $P_1, P_2$, the $s$-channel decomposition is not an efficient way to numerically compute mixed four-point functions. To do that, the $t$- and $u$-channel decompositions are much better. Nevertheless, let us numerically compute $s$-channel decompositions for the purposes of checking crossing symmetry, and of confirming the correctness of Eq. (4.11). We again focus on the segment $z = x + 0.4i$ with

$x \in (-0.5, 1.5)$. On this segment, we computed the four-point function $\left\langle V_{0.356}^{D} V_{0.101+0.5i}^{D} V_{\langle 2, -\frac{1}{2} \rangle}^{N} V_{\langle 2, \frac{5}{2} \rangle}^{N} \right\rangle$ at $c = -0.41$, and found an excellent agreement between the three channels, with $8 - 12$ common digits for all values of $x$. Here is a plot of the real and imaginary parts of this four-point function:

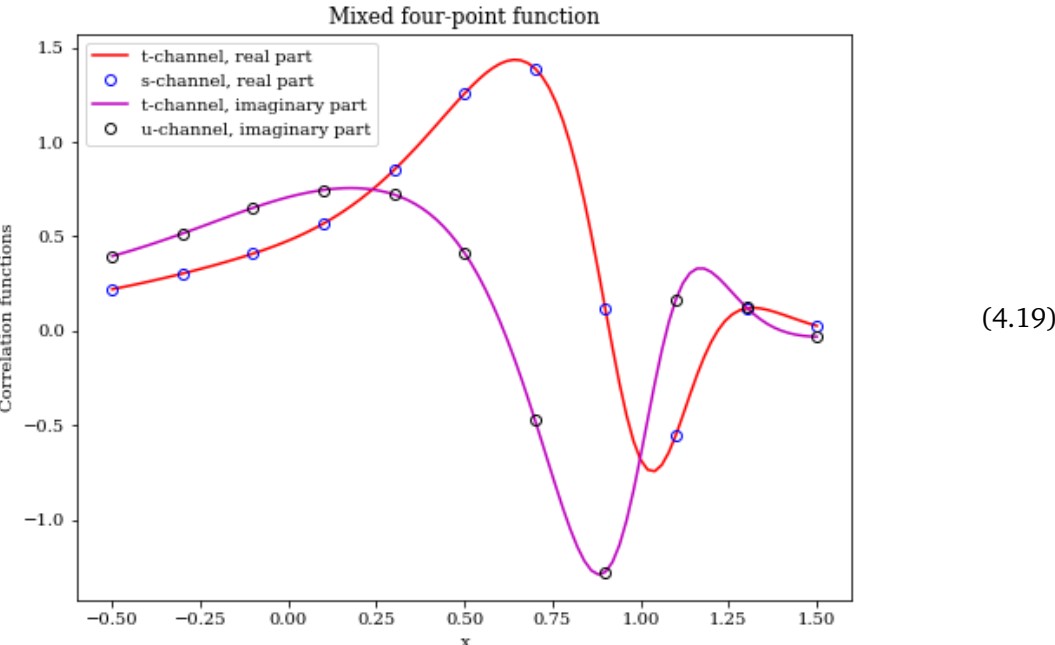

(4.19)

## 4.3 The diagonal three-point structure constant

### Divergent sum and distribution

Let us recast the $s$-channel decomposition (4.16) in the form of Eq. (2.2), i.e. as an expression that involves an $s$-channel spectrum, structure constants, and conformal blocks. Let us formally rewrite the decomposition as

$$\left\langle V_{P_1}^{D} V_{P_2}^{D} V_{\langle r_3, s_3 \rangle}^{N} V_{\langle r_4, s_4 \rangle}^{N} \right\rangle = \beta \int_{\mathbb{R}+i\epsilon} \varphi_{P_1, P_2, P} f(P) dP , \tag{4.20}$$

where the function $f(P)$ is the combination (4.10) of structure constants and conformal blocks, and we define

$$\varphi_{P_1, P_2, P_3} = \sum_{n \in \mathbb{Z}} (-1)^n \frac{\prod_{i=1}^{3} \cos(2\pi n \beta P_i)}{\cos(\pi n \beta^2)} . \tag{4.21}$$

Since the sum over $n$ diverges, this should be considered as a distribution, i.e. as an object that makes sense only in the context of an integral such as Eq. (4.16). The advantage of this formulation is that $\varphi_{P_1, P_2, P_3}$ is manifestly symmetric under permutations of the momentums $P_i$, and can therefore be interpreted as a three-point structure constant (or a factor thereof). This shows that our $s$-channel decomposition is indeed in the form of Eq. (2.2), provided we redefine the diagonal three-point structure constant as

$$\hat{C}_{P_1, P_2, P_3} = \beta \varphi_{P_1, P_2, P_3} C_{P_1, P_2, P_3} , \tag{4.22}$$

where $C_{P_1,P_2,P_3}$ (2.23) is Liouville theory's three-point structure constant. The structure constant $\hat{C}_{P_1,P_2,P_3}$ evades the analytic bootstrap uniqueness result by not depending analytically on the momentums.

**Verlinde formula**

Curiously, the distribution sum $\varphi_{P_1,P_2,P_3}$ can be rewritten in terms of the modular $S$-matrix for Verma modules of the Virasoro algebra,

$$S_{P,P'} = \cos(4\pi P P') . \tag{4.23}$$

We indeed have

$$\varphi_{P_1,P_2,P_3} = \sum_{n\in\mathbb{Z}} \frac{\prod_{i=1}^3 S_{P_{\langle n,0\rangle},P_i}}{S_{P_{\langle n,0\rangle},P_{\langle 1,1\rangle}}} . \tag{4.24}$$

Since $P_{\langle 1,1\rangle}$ is the momentum of the identity field, this is formally identical to the Verlinde formula, where $\varphi_{P_1,P_2,P_3}$ plays the role of fusion multiplicities. In rational conformal field theories, fusion multiplicities are integer numbers: the meaning of having a distribution instead is not clear.

The most mysterious aspect of the Verlinde formula is the summation over momentums of the type $P_{\langle n,0\rangle}$. These momentums do not appear in the non-diagonal sector (2.11) of our theory, and they a priori do not play any special role in the diagonal sector. Studying the boundary theory might shed light on this aspect.

**Further properties**

For special values of the momentums, the three-point structure constant can reduce to a linear combination of Dirac delta functions. The relevant special values are such that a sine factor from the numerator cancels the cosine factor in the denominator. In the notation (2.5) for the momentums, these special values are of the type $P_{\langle 1,s\rangle}$ with $s \in \mathbb{Z}$. Assuming $P_1, P_2 \in \mathbb{R}$, we indeed have

$$\varphi_{P_1,P_2,P_{\langle 1,s\rangle}} = \frac{1}{4\beta} \sum_{s'\in\mathbb{Z}+\frac{s-1}{2}} \sum_{\pm,\pm} \delta\left(\pm P_1 \pm P_2 + s'\beta^{-1}\right) . \tag{4.25}$$

If we now consider the full structure constant $\hat{C}_{P_1,P_2,P_{\langle 1,s\rangle}}$ (4.22), then some of the zeros of the factor $C_{P_1,P_2,P_{\langle 1,s\rangle}}$ (2.23) cancel some of our Dirac delta functions, and we find

$$\hat{C}_{P_1,P_2,P_{\langle 1,s\rangle}} = \frac{1}{4} C_{P_1,P_2,P_{\langle 1,s\rangle}} \sum_{s'=-\frac{s-1}{2}}^{\frac{s-1}{2}} \sum_{\pm,\pm} \delta\left(\pm P_1 \pm P_2 + s'\beta^{-1}\right) , \tag{4.26}$$

where the sum is empty for $s \leq 0$. Now the Dirac delta functions enforce the fusion rule (2.18) of the degenerate field $V_{\langle 1,s\rangle}^D$, thanks to a conspiracy between the smooth and distributional factors of the structure constant. It has long been known that the analytic structure constant $C_{P_1,P_2,P_3}$ does not necessarily enforce the relevant fusion rules when a momentum takes a degenerate value [3, 14]: we now see that the distributional factor restores the fusion rules in some cases.

Let us study how $\varphi_{P_1,P_2,P_3}$ behaves under shifts of the momentums. One shift equation is simple:

$$\varphi_{P_1+\beta^{-1},P_2,P_3} = \varphi_{P_1,P_2,P_3} . \tag{4.27}$$

On the other hand, the shift by $\beta$ is more complicated. Assuming $P_i \in \mathbb{R}$, we find

$$\varphi_{P_1+\frac{\beta}{2},P_2,P_3} + \varphi_{P_1-\frac{\beta}{2},P_2,P_3} = \frac{1}{4\beta} \sum_{s \in \mathbb{Z}+\frac{1}{2}} \sum_{\pm,\pm,\pm} \delta\left(\pm P_1 \pm P_2 \pm P_3 + s\beta^{-1}\right) . \qquad (4.28)$$

Remember that the impossibility of solving the shift equations (2.26) with smooth functions was the reason why we had to take a limit of minimal models. We now find that the distribution $\varphi_{P_1,P_2,P_3}$ solves an analogous equation, which however includes extra terms made of Dirac delta functions.

**How can this be a consistent CFT?**

Having a distributional three-point structure constant is surely exotic, but we know that our mixed four-point functions are very irregular as functions of $\beta^2$ [4], due to the poles of the $t$- and $u$-channel conformal blocks. In our $s$-channel decomposition, the conformal blocks are perfectly smooth, so it is the structure constants that had to be very irregular.

However, another feature of our four-point functions seems to challenge the very axioms of conformal field theory: diagonal four-point functions involve a three-point structure constant that comes straight from Liouville theory, while mixed four-point functions involve another three-point structure constant, which has the extra distributional factor $\varphi_{P_1,P_2,P_3}$. But in a given CFT, the three-point structure constant should not depend on which four-point function we are decomposing. We will therefore have to conclude that the diagonal and mixed four-point functions cannot belong to the same CFT. These two types of four-point functions are both limits of four-point functions of D-series minimal models, but the operations of taking the limit and restricting to the diagonal sector do not commute. As a result, limits of D-series minimal model correlation functions can belong to two different CFTs, depending on the diagonality of the fields. In order to make this point clear, we will study more general multipoint correlation functions.

# 5 Multipoint correlation functions

Since our four-point functions are hard to interpret in terms of a consistent CFT, we now broaden our perspective to multipoint correlation functions. To begin with, let us study whether and how multipoint correlation functions diverge when we take the limit of D-series minimal models.

## 5.1 Divergences in the limit of minimal models

**Influence of diagonal fields**

Let us first consider correlation functions of diagonal fields. We have found that four-point functions of the $(p,q)$ minimal model have a divergence of order $q$ in the limit (2.9), see Eq. (4.4). This divergence comes from the sum over $s$-channel fields in the $s$-channel decomposition. For a $d$-point function, decompositions into conformal blocks involve $d-3$ such sums, and the divergence is of order $q^{d-3}$.

In the presence of non-diagonal fields however, the divergence of correlation functions cannot depend on the number of diagonal fields. This can be seen by considering a decomposition

into conformal blocks such that all diagonal fields fuse with non-diagonal fields:

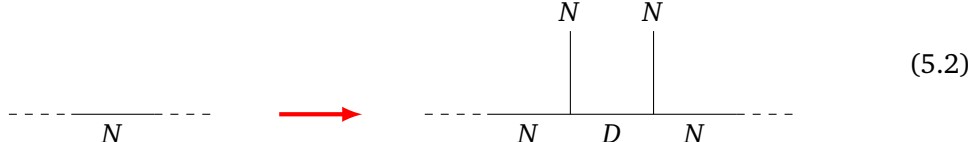

(5.1)

This decomposition relies on the repeated use of the $V^D V^N$ OPE (2.16), which has a finite limit.

**Influence of non-diagonal fields**

It remains to determine how the divergence depends on the number of non-diagonal fields. We already know the behaviour of four-point functions with 2 and 4 non-diagonal fields, see Eqs. (4.9) and (4.5) respectively. This suggests that adding two non-diagonal fields leads to an extra divergence of order $q$. This is most easily seen in decompositions where the extra two fields both fuse with the same non-diagonal field:

(5.2)

The sum over the extra diagonal field leads to an extra divergence of order $q$, for the same reason that four-point functions of non-diagonal fields diverge. To summarize, the behaviour of a correlation function of $d$ diagonal and $n$ non-diagonal fields is

$$
\left\langle \left(V^N\right)^n \left(V^D\right)^d \right\rangle_{\mathrm{MM}_{p,q}} \underset{\substack{p,q\to\infty \\ \frac{p}{q}\to\beta^2}}{\sim}
\begin{cases}
q^{\frac{n}{2}-1} & \text{if } n \geq 2 , \\
q^{d-3} & \text{if } n = 0 .
\end{cases}
\tag{5.3}
$$

**Interpretation**

The diagonal sector (i.e. correlation functions with $n = 0$) behaves differently from the rest of the theory. This implies that we can either define the limit of D-series minimal models as a consistent CFT, or define finite, nontrivial limits for non-diagonal correlation functions, but not both at the same time.

If we insist on having a consistent limit CFT, then the diagonal sector tells us that the diagonal OPE coefficient should be $\lim \frac{1}{q} C_{DD}^D$, where $C_{DD}^D$ is the minimal models' diagonal OPE coefficient. However, in order to decompose a non-diagonal correlation function, we would need an OPE coefficient that remains finite in our limit, rather than diverging as $O(q)$. Therefore, non-diagonal correlation functions are negligible with respect to diagonal correlation functions. The consistent CFT is then reduced to the diagonal sector.

What we actually want is a CFT that contains the finite, nontrivial limits of non-diagonal correlation functions, i.e. $\lim q^{1-\frac{n}{2}} \left\langle \left(V^N\right)^n \left(V^D\right)^d \right\rangle$ with $n \geq 2$. We cannot include the limits of diagonal correlation functions in the same CFT, as these limits would be infinite. Rather, we will define a consistent CFT by completing the limit of the non-diagonal sector. By completing we mean computing diagonal correlation functions using structure constants that are inferred from the non-diagonal sector.

## 5.2 Decomposition into structure constants and conformal blocks

**Limit of the diagonal sector**

Since the diagonal sector does not involve the non-diagonal three-point structure constant, there is no sign problem in this sector. The limit of minimal models straightforwardly leads to Liouville theory, whose three-point structure constant (2.23) is uniquely determined by the analytic bootstrap.

**Ubiquity of distributional three-point structure constants**

We will now argue that as soon as *non-diagonal* fields are present, the *diagonal* three-point structure constant is given by the distributional expression (4.22), in any decomposition of any correlation function.

 This claim may seem implausible at first sight, because the sign problem originated with the *non-diagonal* three-point structure constant. However, we can actually move signs around by renormalizing fields. Schematically, the three-point functions of D-series minimal are of the type

$$\left\langle \prod_{i=1}^{3} V_{\langle r_i, s_i \rangle}^{D} \right\rangle = \text{analytic} \quad , \quad \left\langle V_{\langle r_1, s_1 \rangle}^{D} V_{\langle r_2, s_2 \rangle}^{N} V_{\langle r_3, s_3 \rangle}^{N} \right\rangle = (-1)^{\frac{r_1}{2}} \times \text{analytic} , \qquad (5.4)$$

where "analytic" denotes expression that depend analytically on the diagonal momentums $P_{\langle r_i, s_i \rangle}$. Renormalizing diagonal fields by $V_{\langle r,s \rangle}^{D} \to (-1)^{\frac{r}{2}} V_{\langle r,s \rangle}^{D}$ would make the non-diagonal three-point function analytic, and move the non-analytic sign factor to the diagonal three-point function.

 Let us sketch what happens in the fixed $c$ limit of a $d+2$-point function with 2 non-diagonal and $d$ diagonal fields, $d-1$ of which are degenerate. We consider any decomposition where we start by fusing the two non-diagonal fields with one another:

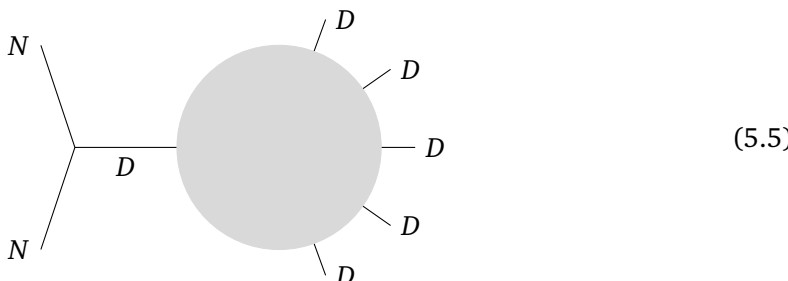

$$(5.5)$$

Whenever we use the degenerate OPE (2.18), we generate a discrete sum over indices $r_j, s_j$. The momentum of the diagonal field that interacts with the non-diagonal fields is of the type $P_0 + \beta \sum_j r_j + \beta^{-1} \sum_j s_j$, and the overall sign factor is therefore $(-1)^{\sum_j r_j} = \prod_j (-1)^{r_j}$, as if each use of the OPE came with its own sign factor. The resulting sums over $r_j, s_j$ are therefore all alternating sum, and they lead to distributional three-point structure constants in the fixed $c$ limit.

**Diagonal sector of the limit theory**

In the limit theory, there should exist correlation functions of diagonal fields. Such correlation functions cannot be computed as limits from minimal models, or as fixed $c$ limits of correlation functions with degenerate fields. Rather, they are defined from their decompositions into conformal blocks, using the distributional three-point structure constant.

In the case of a $d$-point function, the decomposition involves $d-2$ distributional structure constants, and $d-3$ integrals over momentums. A distribution yields a finite result when integrated against a smooth function. Here, conformal blocks play the role of smooth functions, but we have one fewer integral than we have distributions. Therefore, the $d$-point function is still a distribution. In order to make it finite, an extra integral would be needed. For example we could smear one of the $d$ fields, and obtain the finite quantity

$$\int_{\mathbb{R}} dP_1 \, e^{-\lambda P_1^2} \left\langle \prod_{i=1}^{d} V_{P_i}^D \right\rangle_{\text{limit theory}} . \qquad (5.6)$$

The need to smear our correlation functions would make it numerically time-consuming to directly check crossing symmetry of diagonal four-point functions. However, the smearing can also be performed by introducing two non-diagonal fields, bringing us to the non-diagonal sector whose correlation functions are finite. From crossing symmetry in minimal models, we deduce the equality of the two decompositions

$$
\begin{array}{c}
\text{(5.7)}
\end{array}
$$

which implies crossing symmetry in the diagonal sector.

# 6 Conclusion

**Limit of D-series minimal models**

While the diagonal sector of a D-series minimal models is only a submodel of the corresponding A-series minimal model, its momentums still become dense in the real line in the non-rational limit, so that

$$\lim_{\substack{p,q\to\infty \\ \frac{p}{q}\to\beta^2}} \text{MM}_{p,q}^{\text{D-series, diagonal}} = \lim_{\substack{p,q\to\infty \\ \frac{p}{q}\to\beta^2}} \text{MM}_{p,q}^{\text{A-series}} = (\text{Liouville theory})_{\beta^2} . \qquad (6.1)$$

In words, the limits of correlation functions of diagonal fields in D-series minimal models are correlation functions in Liouville theory. It was therefore natural to expect that the non-rational limit of D-series minimal models would be a non-diagonal extension of Liouville theory. Finding this expectation wrong was a major surprise. As soon as some non-diagonal fields are involved, the limits of correlation functions belong to a different CFT, whose diagonal sector differs from Liouville theory, and whose diagonal correlation functions depend on momentums as distributions. Let us call that theory the "limit CFT", although this term does not apply to the diagonal sector:

$$\lim_{\substack{p,q\to\infty \\ \frac{p}{q}\to\beta^2}} \text{MM}_{p,q}^{\text{D-series, non-diagonal}} = (\text{Limit CFT})_{\beta^2}^{\text{non-diagonal}} , \qquad (6.2)$$

$$(\text{Limit CFT})_{\beta^2}^{\text{diagonal}} \neq (\text{Liouville theory})_{\beta^2} . \qquad (6.3)$$

At the level of correlation functions, our limit is finite provided appropriate $q$-dependent prefactors are included, see Eq. (5.3).

**Spectrum, OPEs and structure constants of the limit CFT**

The spectrum of the limit CFT is the limit of the spectrum of D-series minimal models. Collecting Eqs. (2.11) and (2.14), we have

$$\mathcal{S}_{\beta^2}^{\text{Limit CFT}} = \int_{\mathbb{R}_+} dP \ |\mathcal{V}_P|^2 \oplus \frac{1}{2} \bigoplus_{(r,s)\in 2\mathbb{Z}\times(\mathbb{Z}+\frac{1}{2})} \mathcal{V}_{P_{\langle r,s\rangle}} \otimes \bar{\mathcal{V}}_{P_{\langle -r,s\rangle}} \ . \qquad (6.4)$$

The only fusion rule that constrains the OPEs (2.15)-(2.17) is the conservation of diagonality. From its spectrum and fusion rules, the limit CFT therefore looks like a non-diagonal extension of Liouville theory. But its diagonal structure constant $\hat{C}_{P_1,P_2,P_3}$ differs from that of Liouville theory by the distribution factor $\varphi_{P_1,P_2,P_3}$ (4.21). The three-point structure constants of the limit CFT are:

| #non-diagonal fields | Name | Notation | Formula | |
|:---:|:---:|:---:|:---:|:---:|
| 0 | diagonal | $\hat{C}_{P_1,P_2,P_3}$ | (4.22) | (6.5) |
| 2 | non-diagonal | $\frac{C_{P_1,\langle r_2,s_2\rangle,\langle r_3,s_3\rangle}}{\sigma(P_1)}$ | (2.25) | |

**Dependence on the central charge**

The limit of minimal models gives us access to central charges $c \in (-\infty, 1)$, equivalently $\beta^2 \in \mathbb{R}_{>0}$. On this half-line, there is a subset of measure zero where the limit is ill-defined, and this subset includes $\beta^2 \in \mathbb{Q}_{>0}$. The limit CFT actually depends on $\beta^2$ not $c$: since $c(\beta) = c(\beta^{-1})$, there are two distinct limit CFTs for any allowed central charge $c \neq 1$.

From the study of mixed four-point functions $\langle V^D V^D V^N V^N \rangle$ in the $t$-channel, we expect that the limit CFT actually exists on the half-plane $\{\Re c < 13\}$, equivalently $\{\Re \beta^2 > 0\}$, i.e. the values such that the non-diagonal sector's total conformal dimensions (2.13) reach $+\infty$ in real part. Actually, this half-plane is also where the $s$-channel decomposition (4.16) converges, see Eq. (4.18). However, that $s$-channel decomposition has no reason to be valid beyond $c \in (-\infty, 1)$, for the same reason that Liouville theory is not analytic in $c$ near $c \in (-\infty, 1)$ [13]: when $c$ moves away from the real line, infinitely many poles of conformal blocks cross the $s$-channel decomposition's integration line. In particular, we do not expect the diagonal structure constant to be valid beyond $c \in (-\infty, 1)$.

The non-diagonal four-point functions $\langle V^N V^N V^N V^N \rangle$ are probably the easiest to understand for complex central charges, thanks to their coincidence with Liouville theory four-point functions (4.2) in special cases. Let us assume that this coincidence still holds beyond $c \in (-\infty, 1)$: then non-diagonal four-point functions depend analytically on the central charge for $c \in \mathbb{C} - (-\infty, 1)$. To compute them, we cannot simply use the diagonal OPE (2.15): in Liouville theory, this OPE is valid in four-point functions $\langle \prod_{i=1}^4 V_{P_i} \rangle$ with real momentums $P_i$, but acquires extra discrete terms if $P_i$ strays too far from the real line, as typically happens in Eq. (4.2). The discrete terms can then be derived by analytic continuation in $P_i$ [3].

For non-diagonal four-point functions with no relation to Liouville theory, the natural guess is that we also have discrete terms in the $s$-channel decomposition. However, we do not know how to derive these terms. The best we can do at the moment is to guess the discrete terms and numerically check whether the resulting four-point functions are crossing-symmetric. So far, we were only able to do this in four-point functions of the type $\left\langle V_{\langle r_1,s_1\rangle}^N V_{\langle r_1,-s_1\rangle}^N V_{\langle r_2,s_2\rangle}^N V_{\langle r_2,-s_2\rangle}^N \right\rangle$.

**Outlook**

Apart from solving the limit CFT on the whole half-plane $\{\Re c < 13\}$, interesting open problems include studying the limit of D-series minimal model on the torus, disc and cylinder. In particular, the boundary CFT might make sense of the Verlinde formula for the diagonal three-point structure constant. It would also be interesting to understand how the limit CFT behaves in rational limits $\beta^2 \to \frac{p}{q}$, where we already know that we recover minimal model correlation functions in some cases only [4].

Our approach could be generalized to other families of exactly solvable CFTs whose central charges are dense in a line. This includes minimal models for extended symmetry algebras, and fermionic minimal models [15]. The limit of the fermionic minimal model's non-diagonal sector is of the type of Eq. (2.11) with however $(r,s) \in \mathbb{Z} \times (\mathbb{Z} + \frac{1}{2})$ instead of $(r,s) \in 2\mathbb{Z} \times (\mathbb{Z} + \frac{1}{2})$. This allows the spin $rs$ to take half-integer values, and we should obtain a non-rational fermionic CFT.

We dare not suggest looking for applications of the limit CFT. Its dependence on $\beta^2 \in \mathbb{R}_{>0}$ is very singular, and differs from the smooth dependence that we expect in critical statistical systems. This difference was even used for distinguishing the limit CFT from the critical Potts model, although they looked identical from the point of view of the numerical behaviour of certain correlation functions [8, 16].

Ultimately, the limit CFT's singularities follow from the analytic bootstrap's axiom that there exist two independent degenerate fields. In a less singular CFT, we should probably have at most one degenerate field. We may still derive some analytic relations between structure constants [17] or use numerical bootstrap techniques [18], but a full analytic solution of the CFT would be challenging.

## Acknowledgements

I am grateful to Jean-François Bony for decisive help with computing limits of alternating sums, and for pointing out that they do not converge for all irrational $\beta^2$. I wish to thank Ingo Runkel for comments on a draft version of this article, which led to important clarifications and improvements. I am grateful to the anonymous SciPost reviewer for comments and suggestions.

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
