# Peer review of "The non-rational limit of D-series minimal models"

_SciPost Physics Core, doi:SciPost Phys. Core 3, 002 (2020)_

## Round 1 · Referee Report · Anonymous · 2019-12-16

Strengths

1- The paper makes a serious attempt to understand the limit(s) of the D-series minimal models. Not all the results are rigorous, but I did not find any mistakes.

Weaknesses

1- The analysis depends strongly on the fields being clearly divided into diagonal and non-diagonal, but the "non-diagonal" sector includes diagonal fields of the sort $(r,s),(-r,s) = (0,s),(0,s)$. This means that further analysis is required to justify equations (2.16), (2.18), (2.19) etc. I think this could be done, for example it is noted before equation (2.27) that these can be distinguished by their 3 point functions, but at the very least the words "non-diagonal sector" are misleading. This needs to be explained clearly.

2- I though the discussion in in the conclusion was a little disingenuous. The D-series diagonal fields have an identical set of structure constants and correlation functions with a subset of the A-series models, but the A-series models have extra diagonal fields and so it is not especially surprising if the structure constants of the limits are different.

3- I do not know in what sense the degenerate fields "exist" in the theory if they are excluded from the spectrum. They are outside the theory and I would like some justification why they should be able to be included consistently and that deductions from their correlation functions are still valid.

Report

This is an interesting paper adding to the study of limit CFTs and CFTs with continuous spectrum. There are some clear issues involved with defining the limit CFT(s) of the D-models which are discussed and tackled.

I would have liked a clearer demonstrations that there are in fact two different CFTs in the limit, one from $\beta$ and one from $1/\beta$. It is asserted a couple of times but I did not find a clear explanation: apologies if I just missed it.

Requested changes

1- please clarify that the "non-diagonal sector" also includes diagonal fields and that the restrictions on fusion rules etc follow from other considerations than "diagonality".

---

## Round 2 · Referee Report · Anonymous (Referee 1) · 2020-4-28

Report
This is certainly a very stimulating paper. I have spent some time thinking about it and have found it hard to come to a conclusion for reasons below.
I finds it hard to accept one of the main conclusions, which is "We will therefore have to conclude that the diagonal and mixed four-point functions cannot belong to the same CFT". If the limiting procedure results in a CFT then it is a CFT; if it does not, then what is it?
The main problem appears to be the non-analyticity in $\beta^2$ arising from the shift $\sigma(P)$ in equation (2.25). I was for some time misled by this formula which appears to suggest that $\sigma(P)$ is defined independently of $\{r_2,s_2,r_3,s_3\}$. However, since the sign has to change to accommodate changes in sign of three point functions of fields with spin, this cannot be true - for some pairs of $(r_2,s_2)$ and $(r_3,s_3)$, the three point function has to change sign and so $\sigma(P)$ does depend on $\{r_2,s_2,r_3,s_3\}$.
Unfortunately, I cannot find a formula for $\sigma(P)$, or for $C_{P_1,{r_2,s_2},{r_3,_3}}$ itself. It says in [5] that the shift equations are solved (for a sign $f_{2,3}(P)$) but I could not find that solution in that paper, which may very well just be my fault. As it is, the absence of a clear formula for $C_{P_1,{r_2,s_2},{r_3,_3}}$ makes it hard to check whether there are other ways to fix the problem of non-analyticity.
I could imagine that the sign $\sigma(P)$ might have a component (like a cocyle) that depends on $\{r_2,s_2,r_3,s_3\}$ and part that depends on $P$ in such a way that one could split the fields into two sets, one for two possible signs, and for which new fields then have structure constants that are analytic, up to some signs required because of the presence of fields with spin - but it is hard to start thinking about this in the absence of a formula for $C_{P_1,{r_2,s_2},{r_3,_3}}$.
So, to summarise, I have found the ideas very stimulating but have been unable to decide what I think of the results presented here. They seem to me outside what I regard as "likely" and I have not been able to do the sorts of tests of the ideas I would like to do because I have not been able to find the required formulae. This could just be my incompetence, for which I apologise.
If the author can provide a formula for $\sigma(P)$, either explicitly, or a clear indication where it is to be found, then I think that would be a substantial improvement and at that stage I would be very likely to recommend publication. Even if the eventual conclusions are wrong, the paper is still very interesting and stimulating.
I am grateful for the reviewer's work and interest. On the function $\sigma(P)$ and on the "hard to accept" conclusion, I realize that the article is not clear and explicit enough. I would be happy to submit a revised version where I would attempt to clarify these points.

Sylvain Ribault on 2020-05-11 [id 820]
PS: here are the changes in the arxiv version 3:
1.I have written more details on the sign factor $\sigma(P)$. To begin with, I have explained how the shift equations (2.26) are deduced from ref. [5], and why they do not depend on $r_i,s_i$ in our case. Then, I have written explicit expressions (2.29) and (2.30)
for the solutions of the shift equations. Moreover, in order to better show how the signs appear in four-point functions, I have added the diagrams (2.32).
2.In order to clarify the statement that "the diagonal and mixed four-point functions cannot belong to the same CFT", I have added a few lines of explanation after this statement, at the very end of Section 4. I have also added explanations at the end of Section 5.1.
Sylvain Ribault on 2020-05-11 [id 819]
In order to clarify the points raised by the reviewer, I have sent a revised version to arXiv: https://arxiv.org/abs/1909.10784 .

---

## Round 2 · Author Response

List of changes
In response to the reviewer's suggestions:
-
I have written further explanations on the non-diagonal sector, and related it to the model's $\mathbb{Z}_2$ symmetry as discussed in Ref. [10]. See page 5 after Eq. (2.8), and page 6 the beginning of the last paragraph.
-
Writing that the limit of D-series minimal models was expected to be a diagonal extension of Liouville theory was actually an understatement of the major surprise that occurred. I have tried to explain this in more detail at the beginning of the Conclusion.
-
The role of the degenerate fields is now elaborated in more detail at the beginning of Section 2.2 (second paragraph).
-
The lack of symmetry under $\beta \to \frac{1}{\beta}$ is now demonstrated after Eq. (2.13).
Additional changes:
- In the Conclusion (Outlook part), I have added a paragraph on possible generalizations.

---

## Round 2 · List of Changes

In response to the reviewer's suggestions:
-
I have written further explanations on the non-diagonal sector, and related it to the model's $\mathbb{Z}_2$ symmetry as discussed in Ref. [10]. See page 5 after Eq. (2.8), and page 6 the beginning of the last paragraph.
-
Writing that the limit of D-series minimal models was expected to be a diagonal extension of Liouville theory was actually an understatement of the major surprise that occurred. I have tried to explain this in more detail at the beginning of the Conclusion.
-
The role of the degenerate fields is now elaborated in more detail at the beginning of Section 2.2 (second paragraph).
-
The lack of symmetry under $\beta \to \frac{1}{\beta}$ is now demonstrated after Eq. (2.13).
Additional changes:
- In the Conclusion (Outlook part), I have added a paragraph on possible generalizations.

---

## Round 3 · Referee Report · Anonymous · 2020-6-14

Report

Again, I would like to thank the author for their changes. The comments on the limit process require more thought, but they present a point of view that I think is definitely acceptable.

On the issue of the sign $\sigma(P)$, I am sorry that I was not more explicit as I think that the author has not understand my concerns. I was not worried by the shift equations themselves, whether they depend on $(r_2,s_2)$ and $(r_3,s_3)$, only in the solutions and the actual values of $\sigma(P_1)$.

The sign $\sigma(P)$ has to depend on $(r_2,s_2)$ and $(r_3,s_3)$ since, for example, $C_{P,(2,1/2),(4,1/2)} = - C_{P,(4,1/2),(2,1/2)}$ from the standard result for the three point structure constant of fields with integer spin,
$C_{abc} = (-1)^{S_c+S_a+S_b} C_{acb}$ where $S_a$ is the spin of the field $a$, [see e.g. eqn (2.2.48) in arxiv:1406.4290] and the facts that the spin of the field $V_P$ is $0$, and of $V_{(r,s)}$ is $rs$.
Hence, if we replace $\sigma(P)$ in eqn (2.25) by the more general notation
$\sigma_{P_1,(r_2,s_2),(r_3,s_3)}$, it has to be the case that
$\sigma_{P_1,(2,1/2),(4,1/2)} = - \sigma_{P_1,(4,1/2),(2,1/2)}$.
I would like to know what values the author has given for $\sigma(P_1)=\sigma_{P_1,(r_2,s_2),(r_3,s_3)}$ to understand if there is any way to undo the non-analyticity by field redefinitions, splitting the fields into two sets, etc, or (as is obviously suggested) there is none.
I really think that the author should provide the values of $\sigma_{P_1,(r_2,s_2),(r_3,s_3)}$ so that one could check the numerical calculations.

I am sorry that such a seemingly small point should hold up publication, but it seems essential to me to allow readers to reproduce the calculations and to decide for themselves on the possibility of an analytic solution or not.

  • validity: -
  • significance: -
  • originality: -
  • clarity: -
  • formatting: -
  • grammar: -

Author:  Sylvain Ribault  on 2020-06-15  [id 854]

(in reply to Report 1 on 2020-06-14)
Category:
answer to question

In Eq. (2.25) for the three-point structure constant, in addition to $\sigma(P)$, there should be a prefactor $(-1)^{r_2s_3}$. This prefactor obeys $(-1)^{r_2s_3} = (-1)^{r_2s_2+r_3s_3} (-1)^{r_3s_2}$, because $r_1+r_2\in 2\mathbb{Z}$ and $s_2+s_3\in\mathbb{Z}$. So it leads to the expected behaviour of the three-point structure constant when exchanging the fields $2$ and $3$.

Does this answer the question? Please object if it does not. If I receive no objection within a few days, I plan to submit a revised version with the additional prefactor.

Author:  Sylvain Ribault  on 2020-06-24  [id 864]

(in reply to Sylvain Ribault on 2020-06-15 [id 854])
Category:
answer to question

Yes, this sign factor is described in the notebook Correlators.ipynb in the second text cell, and implemented by the line "product = (-1)**sum(fields[i].indices[0]*fields[i+1].indices[1] for i in range(3))" in the method three_shift() of the class FourPoint.

Anonymous on 2020-06-23  [id 863]

(in reply to Sylvain Ribault on 2020-06-15 [id 854])

Just to confirm, is this the value of the structure constants used in the numerical checks?

---

## Round 3 · Author Response

This resubmitted version is on arXiv since May 8th, I am now formally submitting it at the Editor's request.

I am grateful for the reviewer's work and interest. On the function σ(P) and on the "hard to accept" conclusion, I realize that the article is not clear and explicit enough. The resubmitted version addresses these shortcomings.

---

## Round 3 · List of Changes

1. I have written more details on the sign factor σ(P). To begin with, I have explained how the shift equations (2.26) are deduced from ref. [5], and why they do not depend on $r_i,s_i$ in our case. Then, I have written explicit expressions (2.29) and (2.30) for the solutions of the shift equations. Moreover, in order to better show how the signs appear in four-point functions, I have added the diagrams (2.32).

2. In order to clarify the statement that "the diagonal and mixed four-point functions cannot belong to the same CFT", I have added a few lines of explanation after this statement, at the very end of Section 4. I have also added explanations at the end of Section 5.1.

---

## Round 4 · Referee Report · Anonymous (Referee 1) · 2020-7-19

Report
I would like yet again to thank the author for clarifying the results in this paper. I still find the ultimate analysis unconvincing, but the calculations presented are still very interesting and are now sufficiently detailed to allow a reader to repeat the derivations and decide for themselves whether there is an alternative explanation, and I am happy to recommend publication.

Sylvain Ribault on 2020-06-24 [id 865]
To be more accurate: when testing crossing symmetry in Section 4.2, the sign prefactor $(-1)^{r_2s_3}$ of the three-point structure constant does appear. But this sign is not included in the definition of $\sigma(P)$ in version 4 of the submitted article. The factor $\sigma(P)$ itself is taken to be one in the $t$- and $u$-channel calculations, where it plays no role as we do not integrate over continuous momentums. In the $s$-channel calculation that factor is transformed into the distribution (4.11) by taking the limit.

---

## Round 4 · Author Response

I have corrected the overall sign in Eq. (2.25), so that the three-point structure constant has the right behaviour under permutations of the non-diagonal fields. The values of $\sigma(P)$ are still given by Eqs. (2.29) and (2.30), where the constant prefactor can now be set to one, i.e. it is not just $P$-independent.
Let me emphasize that only the $P$-dependence of $\sigma(P)$ matters for taking the limit of a given four-point function. And $\sigma(P)$ does not appear in the numerical tests of crossing symmetry in Section 4.2, as taking the continuum limit transforms $\sigma(P)$ into the distribution (4.11).
Let me emphasize that only the $P$-dependence of $\sigma(P)$ matters for taking the limit of a given four-point function. And $\sigma(P)$ does not appear in the numerical tests of crossing symmetry in Section 4.2, as taking the continuum limit transforms $\sigma(P)$ into the distribution (4.11).

---

## Editorial Decision

published